

# Anthropogenic wetlands due to over-irrigation of desert areas; A challenging hydrogeological investigation with extensive geophysical input from TEM and MRS measurements

Ahmad A. Behroozmand[1], Pietro Teatini[2,3], Jesper B. Pedersen[4], Esben Auken[4], Omar Tosatto[5], and Anders V. Christiansen[4]

5  [1] Department of Geophysics, Stanford University, CA, USA
   [2] Dept. of Civil, Environmental and Architectural Engineering, University of Padova, Italy
   [3] Consorzio Universitario per la Ricerca Socioeconomica e per l'Ambiente (CURSA), Roma, Italy
   [4] Department of Geoscience, Aarhus University, Denmark
   [5] M³E S.r.l., Padova, Italy

10  *Correspondence to*: Ahmad A. Behroozmand (abehrooz@stanford.edu)





**Abstract.** During the last century, many large irrigation projects were carried out in arid lands worldwide. Despite a tremendous increase in food production, a common problem when characterizing these zones is land degradation in the form of waterlogging. A clear example of this phenomenon is in the Nubariya depression, Western Desert, Egypt. Following the

reclamation of desert lands for agricultural production, an artificial brackish and contaminated pond started to develop in the late 1990s, which at present extends for about 2.5 km$^2$. Available data provide evidence of a simultaneous general deterioration of the groundwater system. With the main objectives of (1) understanding the hydrological evolution of the area; (2) characterizing the hydrogeological setting; (3) developing scenarios of artificial aquifer remediation and recharge, an extensive hydrogeophysical investigation was carried out in this challenging environment using Magnetic Resonance

Sounding and ground-based Time-domain EM techniques. The integrated interpretation of the geophysical surveys provided the hydrogeological picture of the upper 100m sedimentary setting, in terms of both lithological distribution and groundwater quality. The information is then used to setup (1) a regional groundwater flow and (2) a local density-dependent flow and transport numerical model to reproduce the evolution of the aquifer system and develop a few scenarios of artificial aquifer recharge using the treated waters provided by a nearby waste-water treatment plant. The research outcomes point to

the hydrological challenges that emerge for an effective management of water resources in reclaimed desert areas and they highlight the effectiveness of using advanced geophysical and modeling methodologies.

## 1. Introduction

The rise of groundwater due to farmland over-irrigation, i.e waterlogging, is a widespread occurrence worldwide, with a large global impact on the available water resources and often contrasting effects on the water quantity, which increases, and

quality, which deteriorates (Scanlon et al., 2007). Waterlogging is due to a number of causes including large percolation losses from the irrigation water applied to the fields and seepage losses from channels providing significant localized sources of recharge water. One example of groundwater level rises in semi-arid regions, due to man-made activities, can be found in parts of the Xinjiang Province, Western China, where the diversion of river water for irrigation rose the groundwater levels from more than 7 m below the ground surface in the 1950s and about 1 m below the ground surface in the late 1980s (Sheng

and Xiuling, 2001). Another example is the Thar desert of India, running along the western border to Pakistan, where a mean water table rise of 1.1 m/year was recorded following the construction of the Indira Gandhi Nahar Pariyojana (IGNP) irrigation system. This system was started in 1960 and an area of nearly 3960 km$^2$ became liable to waterlogging at the end of 2001 (Choubey, 1997; Sharma, 2001). A third example is the farmland along the Lower Arkansas River Basin of Colorado that has been continuously irrigated since the 1870s and began to develop shallow water tables by the early part of

the 20th century, with an average water table depth less than 2 m below the surface in 1999 (Gates et al., 2002). Waterlogging is usually accompanied by salinization of the surface and subsurface environments (Leaney et al., 2003).

When over-irrigation is combined with low-permeable soils relatively close to the ground surface, low relief or depressions, and the absence of natural drainage either through surface water or groundwater systems, waterlogging can evolve into anthropogenic perennial in-land wetlands even in arid or semi-arid regions. These "Arid wetlands", i.e. natural

humid zones in an arid or semiarid climate, which may seem to be a contradiction (Lemly et al., 1993), have been the focus of hydrological research over the last few years because they represent regions of high conservation value and are crucial refuges for native wildlife (fish, amphibians, snails, and plants) and provide a habitat for migratory birds. Worldwide arid wetlands are threatened by increasing anthropogenic pressure leading to water contamination through the development of agriculture in the surroundings and/or water shortages due to surface water/groundwater use caused by the population growth

(Ashley et al., 2004; Li et al., 2015; Minckley et al., 2013).

An unequivocal contradiction arises when wetlands are generated and grow in desert areas due to over-use of water for irrigation. In such cases, the wetlands reflect the consequence of an extremely inefficient water management, with a potential





waste of a precious resource. Egypt is one of the countries where this occurrence is becoming widespread. Since the 1952
revolution, Egypt has tried to increase its agricultural area through the reclamation of desert land. Land reclamation in the
Egyptian context means converting desert areas to agricultural land and rural settlements primarily by "adding water". The
canals fed by the Nile River are extended through existing agricultural areas to supplement the new reclamation zones
(Adriansen, 2009). Hundreds of deep wells have been drilled to tap the Nubian Sandstone aquifer in order to support
agricultural megaprojects developed within the depressions in the middle of the Western Desert. Excessive pumping,
accompanied by a lack of catchment hydrogeological and geomorphological studies, has caused the formation of large
surficial ponds. An example of this occurs in the Baharia depression where El Bastawesy et al. (2013) have shown by remote
sensing analyses that an increase of cultivated land from 40.9 km$^2$ in 1984 to 95.5 km$^2$ in 2011 corresponded with a wetland
enlargement from 0.3 to 4.7 km$^2$. A similar situation has occurred in the El Fayoum depression, located about 100 km south
of Cairo, which is one of the most important agricultural areas in Egypt. The development of the new reclaimed areas located
in the desert land on the boundaries of older cultivated zones has been associated with excessive irrigation, seepage losses
from canals carrying the Nile water to the farmland, ineffective sub-surface drainage, and lack of proper land development.
The movement of seepage water into local depressions has resulted in the accumulation of surface ponds, particularly in the
cultivated low-lying areas, resulting in damage to historical places such as the Hawwara and Lahun pyramids (El Abd and El
Osta, 2014). Using remotely sensed multi-temporal data, similar occurrences have been highlighted by Arnous and Green
(2015) in new reclaimed lands to the east of the Nile delta, a few tens of kilometers from the Suez Canal, with wetlands in
low-lying areas being created by the seepage of water from adjacent irrigated uplands.

In this paper, the attention is focused on the Nubariya depression (30°41' North, 29°52' East) located on the margin of the
Western Desert, in the proximity of the western margin of the Nile delta (Figure 1). The hydrological evolution of this zone
is exemplificative of the other reclaimed desert areas that have experienced over-irrigation. A large portion of the Nubariya
zone has been converted from desert to irrigated farmland since the 1980s. An extensive network of irrigation canals has
been constructed, transporting water from the Nile River. Over-irrigation and seepage from the bottom of the reclamation
canals has significantly raised the groundwater level, resulting in a surface water accumulation in the most low-lying zone
since the early 2000s. Sharaky et al. (2008) noted that a worsening groundwater quality has accompanied the growth of
wetlands in this area.

In the framework of the EU-SWIM IMPROWARE (Innovative Means to PROtect WAter REsources in the
Mediterranean coastal areas through re-injection of treated water - http://www.improware.eu/) project aimed at evaluating
the possible reuse of treated wastewaters (TWW) in this area, an extensive hydrological and hydrogeophysical study was
performed in 2013-2014. With the goals of explaining the groundwater evolution in the depression and proposing a few
scenarios for TWW reuse to reduce the aquifer salinization, a hydrogeological investigation with extensive geophysical input
and numerical modeling was undertaken. The effectiveness of coupling hydrogeophysical techniques with numerical models
in hydrological applications has been demonstrated in recent years over multiple scales, from lab to catchment (Binley et al.,
2015). For instance, controlled infiltration experiments were monitored using time-lapse (TL) electrical resistivity
tomography (ERT), cross-hole TL-ERT, and TL ground-penetrating radar (GPR), with their outcomes properly assimilated
in partially saturated groundwater flow models (Busch et al., 2013; Rossi et al., 2015). TL-ERT and airborne electromagnetic
(EM) surveys were coupled to variable density groundwater flow models to investigate the fresh- saltwater mixing in flat
coastal areas (Comte et al., 2010; Teatini et al., 2010). Airborne EM data were used to distribute hydraulic conductivity of
catchment-scale groundwater flow models (Dickson et al., 2014; Marker et al., 2015).

Here, based on our previous experience in use of hydrogeophysical methods for groundwater characterization
(Behroozmand et al., 2013; Vilhelmsen et al., 2014), a joint magnetic resonance sounding (MRS) and ground-based transient
EM (TEM) survey was performed on a ~9km$^2$ zone at the southwestern side of the Nubariya pond. The TEM method
measures electrical resistivity of the subsurface, which can be linked to water quality and soil type, in a three-dimensional





(3D) setting (Foged et al., 2014). The MRS method is used to measure free water content (i.e. total volume of water that can freely move and is not bound to the grain surface) directly and provides information on pore size distribution (Legchenko et al., 2004). The free water content is measured in percentage, and hence, in the case of a saturated porous media the free water content measurement indicates effective porosity. By combining both methods a detailed hydrogeological model of the investigated area (see Sect. 2 for details) in terms of hydrogeological structure and water quality was built down to about 100 m depth, i.e. the maximum depth of investigation.

With the main goal of reconstructing the past multiyear hydrological evolution of the area driven by the need for reclamation and its possible future behavior in relation to artificial aquifer recharge using TWW, two numerical models were developed: a 3D groundwater flow model for the entire depression area (25×35 km) and a 3D density-dependent groundwater flow and transport model for a smaller zone (2.7×2.9 km) in the southwestern side of the ponded area caused by over-irrigation (see Sect. 2 for details). The models were constrained by using the results of the geophysical survey, an interpretation of pump tests, a digital elevation model of the depression surroundings, and information on the evolution of the pond and the irrigated area derived from satellite images acquired since 1984. The local model is nested into the depression-scale model in the sense that the results from the latter in term of piezometric head are used as boundary conditions in the former.

In developing countries and many other parts of the world these reclaimed-desert environments are generally quite challenging to access and investigate. Moreover, the development of anthropogenic wetlands as a byproduct of over-irrigation is a process generally not well investigated in the hydrological community. Here, we present a unique case study where the evidence of the occurrence is profound. Finally, from a more technical point of view, the advantages of integrating MRS and TEM methods to discriminate between different deposits and groundwater quality are clearly highlighted for the first time, to our knowledge, in wetland environments.

The paper begins with a section that describes the environmental setting and hydrological evolution of the study area. Secondly, the MRS and TEM methods are briefly introduced, the results from surveys carried out in Nubariya are summarized, and a hydrogeological model obtained by the joint inversion method is shown. The set-up of the depression-scale and local-scale numerical models is presented and the simulation outcomes are shown. The final sections discuss the value of the proposed approach and draws the conclusions.

## 2. The Nubariya depression: evolution and challenges

The study area is located in a complex sedimentary setting along the transition between the Quaternary deposits of the Nile River delta to the west and the Pliocene and Miocene of the Western Desert to the east (Figure 1). The area is characterized by low relief with elevation from 0 to 100 m above mean sea level (amsl). The main land forms developed through the interaction of the geological structures, the processes of wind and surface water erosion, and the climatic conditions (Dawoud et al., 2005; Hassan et al., 2012; RIGW and IWACO, 1991). Hydrogeological investigations at the regional scale highlight the presence of loose coarse Miocene sands with clay lenses in the upper 200 m overlain by Pliocene deposits consisting mainly of estuarine clayey facies at the base and fluvio-marine and shallow marine limestones at the top where they are exposed in the lowest parts of the landscape and their vicinities. Quaternary deltaic sediments from sands to silt dominate in the eastern part of the study area. The Quaternary unit gradually decreases in thickness from the Rosetta branch of the Nile River, to almost zero along the linear depression from Nubariya to the north and Wadi El-Natrun southward, where they interdigitate more or less sharply with the Pliocene unit.

The Nubariya depression represents the northernmost tip of the Wadi El-Natrun elongated depression situated approximately 50 km to the south. Unlike the latter, which has been extensively investigated from the hydrological, hydrogeological, and environmental perspective over the last few decades due to the presence of alkaline lakes (Atwia et al.,





2012; Khalil and Santos, 2013), very little hydrogeological information is available on Nubariya even though it has been subject to more recent reclamation and development.

With a minimum elevation of about 7 m amsl, the Nubariya depression is surrounded by sandy hills with a mean
elevation ranging between 30 and 100 m amsl (Figure 2). The contemporary Digital Elevation Model (DEM) of the area has been obtained from the ASTER-GDEM, which was calibrated by a kinematic DGPS in situ survey carried out in 2014. Over the last three decades, large portions of this desert area have been converted into productive farmland. A remote sensing investigation with a series of Landsat images acquired between 1984 and 2014 clearly shows that a pond started to develop in 1999 and expanded significantly up to the 2014 areal extent equal to about 2.5 km$^2$ (Figure 3). The pond enlargement
corresponds to the expansion of the cultivated areas in the surroundings of the depression, providing evidence of a connection between the two (Figure 3). The reclamation efforts in the Nubariya area relied on Nile water diverted through a set of main (Nubariya Canal) and secondary canals, which are usually unlined. The surface water systems in Nubariya, in most cases, cut through sands; therefore, a direct connection between surface and ground water exists in most parts of the irrigation system. Seepage from groundwater to surface water system and vice versa is very common depending on the water
levels. Overall, the Nubariya canal and its main branch (Nasr canal) recharge the groundwater system in the study area. Because of an inadequate network of drainage ditches, excessive irrigation (via surface irrigation techniques), and seepage from the bottom of the reclamation canals, the groundwater level rose and formed a large water accumulation in the depression. The wetland enlargement was followed by a general deterioration of the surface and subsurface environment. Farmhouses and roads were permanently flooded (Figure 4) and the water quality was degraded by evapotranspiration and
the accumulation of salts in the soils associated with shallow water tables, and fertilizer leaching into underlying aquifers that discharge to the pond (**Error! Reference source not found.**). Despite the nature of the pond, coliforms are practically absent (Fadlelmawla, 2014), also due to the high salinity (Dawe and Penrose, 1978).

The artificial pond has grown despite the climatic conditions. Typical climate that prevails in the area is characterized by high temperatures that span from high 30°C in the summer to few degrees above zero on winter nights. Rainfall occurs
during December to February with an average of 100 mm/year recorded in the northwest part of the area. Usually, the relative humidity ranges from 88% over winter, to 40% in summer. The mean monthly value of potential evaporation ranges between 15.4 mm/day in June and 1.6 mm/day in January (Dawoud et al., 2005).

### 3. Hydrogeological and hydrogeophysical characterization

#### 3.1. Hydrogeological survey

Hydrogeological and chemical investigations were carried out to collect information about the link between the subsurface lithostratigraphy and water quality in the pond, thus providing specific ancillary data for the calibration of geophysical measurements.

With this aim, five boreholes (red triangles in Figure 5) were drilled with continuous core sampling, and pumping and recovery tests. Depth of drilling ranges between 25 and 55 m from the ground surface. Moreover, a number of slug tests
were carried out to characterize the infiltration rate, hence the soil type, of the uppermost deposits.

The lithological analyses of the recovered core revealed the presence of a top 10- to 15m thick sandy unit underlain by a 15- to 20m thick clay layer, overlying limestone (fine sand) at the base borehole. The interpretation of the well tests provides an estimate of the limestone hydraulic conductivity $K$ and elastic storage $S$ in the range of $1 \times 10^{-4} – 6 \times 10^{-4}$ m/s and $\sim 8 \times 10^{-3}$, respectively. Slug tests and data in the literature (Dawoud et al., 2005) suggest $K$ values of approximately $1 \times 10^{-4}$ and $1 \times 10^{-7}$
m/s for the top sand and the clayey deposits, respectively.

Results of chemical analyses of groundwater and pond water samples are summarized in **Error! Reference source not found.**. The measurements highlight a deterioration of the groundwater quality, mainly in terms of NaCl concentration.





Moreover, the similar proportional concentrations of various anions and cations in water samples taken in the pond and in the boreholes suggest a significant mixing between the two water bodies.

**3.2. Transient electromagnetic survey**

The transient electromagnetic (TEM) method has been widely used in groundwater studies (Auken et al., 2003; Danielsen et al., 2003; Siemon et al., 2009) and offers a relatively fast and cost-effective way to obtain information about the ground electrical resistivity down to depths of few hundred meters. The ground-based TEM method employs a transmitter loop, deployed on the ground surface, which generates a primary magnetic field. After a short duration (~10 ms), the current

is switched off abruptly (within a few μs) and the secondary rate of change with time of the magnetic fields from the induced eddy currents in the ground is measured using an induction receiver coil on the surface. Hence, the observed data are voltage data as a function of time. The early time data, typically measured from a few μs after the current shut off, primarily reflect the resistivity of the top layers while the late-time data, typically measured until a few ms after current shut off, provide the resistivity information of the deeper layers. These data measured at different times after the current shut off are then

interpreted in terms of the subsurface resistivity as a function of depth. For basics of the TEM method see Christiansen et al. (2006).

**3.2.1. Field survey and measurement setup**

The TEM survey was carried out in December 2013 using the WalkTEM system (Nyboe et al., 2010). The full survey comprised 127 soundings, 110 of which were located on a 200m spacing grid within the 2.7 km × 2.85 km groundwater

model area, whereas the remaining 17 soundings are located outside the grid to investigate regional geological changes (Figure 5). The WalkTEM system was configured in a central loop configuration with a dual-moment setup (low moment and high moment) in order to obtain soundings containing high quality early time measurements using a low magnetic moment and late-time measurements with a high magnetic moment. The two moments cover the entire depth range from the surface down to approximately 150 m. Furthermore, two different setups were used for the survey, one using a 40 m × 40 m

transmitter loop and another using a 100 m × 100 m transmitter loop resulting in a larger moment and hence larger depth of investigation. **Error! Reference source not found.** documents the used WalkTEM system setup. The instrument was calibrated prior to the survey following the approach of Foged et al. (2013).

**3.2.2. Results**

The quality of the collected data is high and appropriate for the planned analyses. For the majority of the measurements

there are usable data points from 5 μs to 10 ms for the 40 m × 40 m loop and to 20 ms for the 100 m × 100 m loop, respectively. Consequently, for most of the measurements the high signal-to-noise ratio required only a limited amount of processing. However, in some parts of the survey area the signal level for the late time gates fell below the noise level and these gates were removed. In the following processing, the data were inverted using the AarhusInv inversion code (Auken et al., 2015) with a multi-layer model setup (here 20 layers) in which the layer boundaries are fixed and the resistivities of the

neighboring layers are tied together via smoothing constraints. In addition to characterization of overall structural model of the subsurface, multi-layer models help assess homogeneity of each unit. Figure 6 shows a west-east cross section through the resistivity structure obtained from the TEM data. The location of the profile is shown as a black line in Figure 5. The color scale is faded white below the depth of investigation, i.e. the depth to which the model can be trusted (Christiansen and Auken, 2012). The results of the TEM survey suggest four distinct layers in the upper 100-150 m. The top layer is not

continuous, vanishing in the easternmost part of the profile. It is characterized by resistivity values greater than 10 Ωm and a maximum thickness of approximately 20 m. Layer 2 is 20-30 m thick with a low resistivity value of 1-4 Ωm. Layer 3 has a





resistivity value of 4-10 Ωm and a thickness of roughly 100 m. The last layer has a resistivity of 1-4 Ωm and its top boundary is located at elevation of -120 m amsl. The 2D cross section shown in Figure 6 reveals the lateral homogeneity of the geological model of the area and defines a stratigraphy as: unsaturated sand (with a resistivity value of 10-200 Ωm),

saturated sand (brackish, with a resistivity value of 4-10 Ωm) and saturated sandy clay (with a resistivity value of 1-4 Ωm). The bars on the plots of Figure 6 represent the MRS results in the vicinity of the profile, as discussed in the following section.

### 3.3. Magnetic resonance sounding survey

Magnetic resonance sounding (MRS; also called surface NMR) is an electromagnetic geophysical method, which
noninvasively measures water content and pore structure. The method works based on the physical principle of nuclear magnetic resonance (NMR) and is directly sensitive to water molecules and their interaction with the pore space. In short, using Earth's magnetic field, resonance excitation occurs by passing a tuned AC current into a large wire loop deployed on ground surface. The corresponding energizing magnetic fields propagate in the subsurface and at any position in the subsurface a component of the energizing field excites water molecules. The measurement continues by turning off the
energizing pulse and recording the responses from all subsurface excited protons as they gradually return to their initial (equilibrium) state. This experiment is called a free induction decay (FID) and is mostly used for MRS data acquisition. The experiment is repeated for a number of energizing pulses at different pulse moments (defined as the product of the current amplitude and the pulse duration) by which different volumes of the subsurface are sampled and therefore depth information is provided. For more information about the principles and application of the MRS method see, e.g., Behroozmand et al.
60 (2015).

In near-surface geophysics, MRS is commonly used to estimate free water content (or porosity in the case of saturated porous media) and hydrogeological properties such as pore size and hydraulic conductivity. The initial amplitude of the MRS decaying signal is proportional to the water content while its relaxation rate provides information about the pore structure. Hence, based on their relaxation time values, the soils can be classified as fine, medium and large materials. A
small relaxation time typically indicates fine-grained material whereas a high relaxation time indicates coarse materials. The method is well established for near-surface characterization (Chalikakis et al., 2008; Günther and Müller-Petke, 2012; Knight et al., 2012). Different studies have also dealt with estimating hydrogeological parameters from MRS and found good correlation between MRS-derived parameters with those from borehole aquifer tests (Boucher et al., 2009; Herckenrath et al., 2012; Plata and Rubio, 2008; Vilhelmsen et al., 2014, 2016; Vouillamoz et al., 2012, 2015).

### 3.3.1. Field survey and measurement setup

The field campaign consisting of 6 MRS soundings was performed in January 2014. The locations of the MRS soundings are shown as green squares in Figure 5. Prior to the survey, a noise scouting study was carried out over the entire area to investigate the ambient-electromagnetic noise condition, and to propose potential locations for acquiring MRS data. The results of our noise scouting represented low noise levels in the area, which suggests a good site for measuring MRS data
using a relatively low number of repeated measurements (free induction decays, FID). We used the NUMIS Poly system (IRIS Instruments) for data acquisition and the system was configured in coincident loop geometry, with a 100m side square loop as both a transmitter and a receiver. In addition, up to two reference loops were deployed for recording noise. **Error! Reference source not found.** summarizes the measurement configuration used in the study. The measurement sequence consisted of a noise record before excitation, an energizing pulse, a measurement dead time (to switch from the transmit to
receive phase), followed by a recorded FID (see **Error! Reference source not found.** for details). We measured FIDs for 16 pulse moment values and a stack size of 30 was more than enough to obtain high-quality data.



### 3.3.2. Results

The data were processed following the approach described by Dalgaard et al. (2012) and Larsen et al. (2013), and inverted jointly with the TEM data following *Behroozmand et al.* (2012a, 2012b). We used the AarhusInv code (Auken et al., 2015) for inversion of the data. As an example, the bars on top of the TEM plots in Figure 6 show results from three MRS soundings in the vicinity of the profile. In the top figure, the MRS bars show the free water content, whereas in the bottom plot they display the relaxation times. By comparing the TEM results with the MRS results at profile distances 180, 850 and 1900 m one can translate the geophysical results into geology. The MRS measurements suggest no free water in the first layer. This, together with a relatively high resistivity of 10-200 Ωm, corresponds to unsaturated sand which thickens to the west. This layer disappears in the eastern part where the profile approaches the pond. The presence and suggested thickness of this layer is confirmed by the other MRS measurements. For instance, at 1900 m along the profile (site MRS03) the MRS measurement indicates almost no water and a very low relaxation time (also see Figure 7). As for the second layer in Figure 6, very low resistivity of 1-4 Ωm corresponds with a sandy clay layer or a sand layer saturated by salty water. The results of the MRS03 measurement confirm the latter (see Figure 6 and Figure 7) as they show relatively high water content and relaxation time values in the layer, indicating brackish water. Additionally, it confirms the presence of a 10m thick clay layer with a low resistivity of 2 Ωm, and lower water content and relaxation time. Below the clay layer, the free water and relaxation time are high again, those indicating brackish water. The three MRS soundings in Figure 6 penetrate into the top part of layer 3 that is approximately 100 m thick and has a resistivity of 4-10 Ωm. The MRS results suggest a relatively high water content and a relaxation time, which corresponds to a sandy layer. Because of the fairly low resistivity of the layer it can be concluded that the aquifer is filled by water with a moderate salt concentration, i.e. the groundwater is brackish. Interpretation of the fourth layer in an elevation of -100 m amsl remains TEM driven because MRS cannot penetrate that deep. We observe another low-resistivity layer with a resistivity of 1-4 Ωm, which indicates another clay layer or an increase in aquifer salinity.

Figure 7 displays the detailed inversion results for sounding MRS03. The MRS and TEM data were inverted jointly. The top row shows the MRS observed data (column 1), simulated data (column 2) and weighted data residuals (column 3), as well as the TEM data fit (column 4). The estimated model fits both datasets very well, as shown in columns 3 and 4. Row 2 shows the inversion results (black lines) in terms of resistivity (column 1), free water content (column 2) and relaxation time (column 3). The gray error bars denote model parameter uncertainties, shown as the 68% confidence intervals. The model is well determined and is in a good agreement with lithological information obtained from a nearby borehole (column 4). It is noteworthy that the MRS results vary along the profile in Figure 6 suggesting the clay content of the second layer may vary.

Overall, The outcomes of the hydrogeophysical surveys integrate satisfactorily with the lithological and hydrological characterization available from the wellbore information.

The derived hydrogeophysical model was used as an input in the hydrological model of the area, as will be discussed in the following section.

### 4. Hydrogeological Modeling

Numerical modeling was developed to gain some fundamental insight into the processes that explain the observed pond development and provide some preliminary information on the possibility of using TWW for managed aquifer recharge (MAR) in the area. Based on the hydrogeophysical characterization of the area, we set up a simplified groundwater flow depression-scale model and a density-dependent flow and transport local-scale model to verify the possible connection between the desert reclamation and the wetland growth and to test the potential and effect of recharging the saline aquifer by fresh TWW, respectively.



The models, particularly the one developed at the local scale to investigate possible MAR scenarios, rely strongly on the outcome of the hydrogeophysical investigations. Both the geometry of the hydrogeological layers (maps of the top and bottom of the various units) and the distribution-versus-depth of the groundwater quality have been derived by integration of the TEM and MRS results.

### 4.1. Evolution of the pond

#### 4.1.1. Model set-up

The objective was to investigate the possibility of a surplus of irrigation water being responsible for the formation and growth of the pond at the boundary of the desert area by modeling the hydrological response of the zone to the reclamation carried out during the period 1984 - 2014. By simulating the decadal rise of water table, the model also aimed to show inefficient use of the available water resources. Furthermore, the results from the regional-scale hydrological model can be used to provide boundary conditions for the local-scale MAR model.

For this purpose we used a 3D Richards' equation solver that is part of the more general code CATHY (Camporese et al., 2010). The model domain is showed in Figure 2 by the black box. The areal extent amounts to 800 km$^2$, i.e. approximately 32 and 25 km in the southwest - northeast and northwest - southeast directions, respectively, and is centered on the depression. Such a large area has been selected in order to limit the effects of the large uncertainties related to the boundary conditions on the zone of main interest, i.e. the pond surroundings. According to the available piezometric information (Dawoud et al., 2005) and the digital elevation model (DEM), the domain orientation has been selected in order to use a no-flow condition along the southwest - northeast (A-B and C-D, Figure 2) lateral boundaries of the model. The contemporary DEM represents the top surface of the model, with a no-flow bottom bound placed at -34 m amsl, i.e. approximately in the middle of the first confined aquifer. Boundary conditions defined on the current land surface and along the A-D and B-C (Figure 2) lateral sides are quite complex. Regarding the ground surface, keeping in mind the multi-decade reference period spanned by the simulations, the following conditions were used:

- a seepage face condition on the depression zone, i.e. for the surface nodes around the pond with an elevation less than 15 m amsl;

- a net specific recharge of 400 mm/yr is uniformly distributed on the reclaimed zone. Following the development of the farmland as defined from remote sensing data (Figure 3), the recharge zone was assumed to increase from 1984 to 2014 as shown in Figure 2. The recharge amount was obtained by combining average estimates of the aquifer recharge by downward percolation of irrigation surplus water (in desert areas, relatively high 365–550 mm/yr leakage rates have been quantified for basin, furrow, and sprinkler irrigation, with much lower 40–190 mm/yr rates for drip and central pivot irrigation (Dawoud et al., 2005)), rainfall infiltration, and exfiltration due to evaporation;

- a null infiltration/exfiltration in the desert zone, i.e. the portion of the domain complementary to the reclamation area. Due to the relatively large depth to the water table and the lack of specific data, a null balance between rainfall and evaporation has been assumed for simplicity.

Fixed and time-dependent Dirichlet conditions were prescribed along the northwest - southeast bounds toward the desert (A-D, Figure 2) and the Nile delta (B-C, Figure 2), respectively. The water table was set to 0 m amsl along A-D, with sensitivity analysis that has demonstrated a negligible influence of the selected value on the piezometric evolution in the area of interest. Information at regional scale reveals that the water table was continuously rising in the Nile deltaic region (El-Sayed et al., 2012) . However, due to the lack of specific data, we have used the rise of the water table along B-C as a calibration parameter in order to match the date of the pond formation and the pond enlargement versus time. This represents the best information available to calibrate the model at the scale of interest. The approach suggested a linear rise from +10 m amsl in 1984 to +17 m amsl in 2014, i.e. to only a few meters below the land surface. These values seem plausible on





account of available regional information (El Molla et al., 2005; Mabrouk et al., 2013) and a few surveys carried out in the area.

The DEM was used to derive the grid discretization in the horizontal direction, with the characteristic element dimension that reduces from 1000 m on the external boundaries to 50 m around the pond to allow for a more accurate representation of the infiltration/exfiltration processes and changes in time and space of the groundwater table and saturation degree. The 3D finite element (FE) grids consists of 146'275 nodes and 830'808 elements, with 24 layers of tetrahedra that are used to
represent the three main lithostratigraphic units defined from the hydrogeological and hydrogeophysical investigations (Figure 8).

The hydrological properties ($K$ and $S$) of the three main hydro-stratigraphic units were assigned according to the values described above, with the parameters of the van-Genuchten retention curves for the top sandy unit derived using the Rosetta code (Schaap et al., 2001) and equal to $\psi_s$ (capillary pressure) = -0.37 m, $n$ = 4, and $S_{wr}$ (residual water saturation) = 0.15.

### 4.1.2. Results

A steady-state simulation was initially performed to fix the initial condition in the whole domain for the transient run. The solution was obtained by prescribing a water table elevation at 10 m and 0 m above msl on the B-C and D-A (Figure 2) boundaries, respectively. The initial distribution of pressure and saturation degree $S_w$ thus obtained can be considered representative of the hydrological condition in 1984, when the area was still totally desert with negligible reclamation (Figure 3a). With the prescribed setting, the depth to the water table is approximately 10 m below the most depressed portion
of the Nubariya area (Figure 9a).

Then, the model was applied over the period between 1984 and 2014. Figure 9 shows the evolution of $S_w$ along the vertical cross-section E-E traced in Figure 2 as computed using the CATHY code. The distribution of the saturation clearly describes changes in the water table, which evolved due to water leakage from the land surface into the aquifer. The recharge
raises the water table in the eastern part of the domain, with the groundwater level that reached the bottom of the depression approximately in 2007-2008. From 2007 to 2014, the water table rose further, completely saturating the lowest portion of the depression. This caused the waterlogging of the most depressed zone and the pond development and growth. A comparison between the model outcome and the pond's extent acquired from satellite images is presented in Figure 10 for the years 2007 (a), 2010 (b), and 2014 (c). The contour line $S_w$ = 0.9 can reasonably be assumed as the boundary of the waterlogged zone.
The modeled evolution of the pond extent satisfactorily matches the remotely sensed information. It is noteworthy that some westward portions of the pond located outside the simulated zone with $S_w$ > 0.9 are waterlogged due to deep excavation for construction material.

### 4.2. Expected evolution linked to MAR scenarios

MAR using treated wastewater is one of the main challenges in Egypt faces with the reduction of freshwater due to
climate change and the growth in water demand due to the population increase and the enlargement of desert areas reclaimed for agricultural purposes. Reuse of TWW (municipal and to some extent industrial wastewater) is considered an effective water saving measure in areas where this water would otherwise be lost from the Nile system. Primary use of TWW is for irrigation of green areas (landscape development) and non-food agriculture, or the improvement of aquifer quality by mitigating or countering saltwater intrusion and contamination. The planned areal extent based on treated wastewater is
some 250'000 feddan (approximately 1'000 km$^2$). Most of the planned area is located in the Western and Eastern Delta, with Greater Cairo and Alexandria as the main suppliers of treated wastewater (United States Agency for International Development - USAID, 2010).

Within this general context, one of the aims of the IMPROWARE project was to perform a preliminary evaluation on the effectiveness of MAR in the Nubariya zone using TWW provided by the Nubariya wastewater treatment plant (WWTP).





Within the framework of IMPROWARE, an existing WWTP located in the pond surroundings has been updated by a constructed wetland (CW) tertiary system in order to improve the quality of the plant effluent to a level compatible with MAR. The CW treatment capability in the present condition amounts to 160 m$^3$/day, with a potential increase up to 6'000 m$^3$/day representing the working WWTP flowrate of domestic wastewater.

### 4.2.1. Model set-up

A number of preliminary simulations were performed at a local scale to understand the effect of recharging the aquifer system in the Nubariya. This local scale coincides with the area where hydrogeophysical surveys have led to an in-depth understanding of the geologic setting. The U.S. Geological Survey SUTRA code (Voss and Provost, 2002), which can handle density-dependent flows under saturated-unsaturated conditions, was used to simulate a number of scenarios of artificial aquifer recharge. Based on the hydrogeological setting of the study area, the limestone unit identified in the depth

range between 0 and -100 m amsl (Figure 8) was selected as the target aquifer for water injection because of its uniform and large thickness, moderate salt contamination, and relatively high hydraulic conductivity. At this preliminary stage of the study, the following major assumptions were used: i) the injected water is fresh compared to the formation water; ii) well clogging is neglected; and iii) the hydrological regime at the model boundary is constant in time and derives from the outcome of the depression-scale model as of 2014.

The model domain extends approximately 2.7 km in the southwest - northeast direction and 2.85 km in the northwest - southeast direction (Figure 2), roughly coincident with  the area characterized by the MRS and TEM surveys at the southwestern tip of the pond. The model spans the depth range from the contemporary ground surface to the bottom of the fine-sand confined aquifer, approximately -100 m amsl. This latter was mapped by interpolating the models derived from the few deep TEM soundings (Figure 11). The domain was characterized by the three main hydro-stratigraphic units, i.e. sandy,

clayey and limestone layers, consistent with the available data and the depression-scale model. The hydrogeological parameters were also assigned accordingly, in agreement with the values used in the regional model. The maps of the depth of the unit interfaces (top and bottom surfaces) were obtained from the TEM and MRS measurements (Figure 11).

The domain was discretized into prismatic elements. A 2D grid composed of 7'077 nodes and 6'965 quad elements, with the dimension ranging between 50 m on the external boundaries and 20 m in the central area where injection is planned, was

developed initially. Then, the 2D FE grid was "projected" vertically to generate a 3D FE mesh for a total of 219'387 nodes and 208'950 elements. A vertical discretization into 30 layers of 0.5- to 15m thick allows for an accurate reconstruction of the geological formations detected in the area (Figure 12).

The distribution of the salt concentration $c$ has initially been assumed constant in each formation, as suggested by the layering provided by the geophysical investigations. The $c$ values have been derived from the analyses on collected water

samples (Table 1). The solute concentration, expressed in term of dissolved mass fraction ($M_s/M$ where $M_s$ is the mass of dissolved components and $M$ is the fluid mass), are the following: $c_{top\ sand}$=0.013 and $c_{limestone}$=0.016 kg$_{salt}$/kg$_{water}$, i.e. 16 g/l. An intermediate value has been assigned to the clay layer and we have supposed $c$=0 in the unsaturated zone. The following boundary conditions were prescribed:

- Null groundwater and concentration flux along the southwest - northeast boundaries (side A'-B' and C'-D' in Figure
2). These bounds have been properly selected along a direction parallel to the main flow direction as emerged from the depression-scale model;

- Dirichlet constant conditions on the northwest - southeast bounds (side A'-D' and B'-C, in Figure 2). The pressure distribution provided by the depression-scale model in 2014 and the initial $c$ values described above were assigned;

- Neumann no-flux condition on the land surface and domain bottom.



### 4.2.2. Results

A steady state simulation was initially performed to acquire an equilibrated condition in terms of pressure and concentration in the whole domain to be used as the initial state for the transient runs. Due to the lack of specific tracer tests, a sensitivity analysis on the hydrodynamic dispersivity (longitudinal dispersivity $\alpha_L$ and transverse dispersivity $\alpha_T$) was also performed, spanning the range $10<\alpha_L<100$ m as suggested by Gelhar et al. (1992) for a characteristic problem dimension on the order of 100 m. The outcomes provided in the following were obtained using $\alpha_L=20$ m, $\alpha_L/\alpha_T=10$, with the extension of the mixing zone $m_L$ between fresh and in situ brackish water in the longitudinal (horizontal) direction that ranges between 280 m ($\alpha_L=10$ m) and 570 m ($\alpha_L=100$ m), with $m_L=340$ m with $\alpha_L=20$ m.

Three major scenarios were used to address the variability on the injection layout: $S_1$) a single well; $S_2$) three wells along an alignment, spaced between $d=60$ m ($S_{2A}$) and 100 m ($S_{2B}$), and with the central well in the same position of the well in $S_1$; and $S_3$) three wells disposed with a radial configuration, with the circle radius varying between $r_d=60$ m ($S_{3A}$) and 100 m ($S_{3B}$) and the center in the same position of the well in $S_1$. A 20m long well intakes are positioned in the lowest portion of the limestone aquifer in order to keep as compact as possible the freshwater volume stored in the subsurface by taking advantage of fate of the lighter injected freshwater with respect to the denser brackish groundwater. The injection rate of $Q_{inj}=6'000$ m³/day is simulated for a period of 2 years. $Q_{inj}$ is evenly distributed between the three wells in scenarios $S_2$ and $S_3$.

The results in term of $c$ distribution at the end of the simulation period are presented in Figure 13 and Figure 14. Figure 13 shows the salt concentration at the top of the injected aquifer for the various scenarios listed above. The evolution of $c$ in time and space on a vertical cross section through the barycenter of the well layout is presented in Figure 14 for the scenario $S_{3A}$. The injected water moves upward to the bottom of the clay layer, which stops the vertical migration, and then spreads horizontally forming a significant volume of freshwater. The freshwater gently flows southwestward due to the natural hydrodynamic regime. The value $d=r_d=60$ m represents the maximum distance between the wells that allows to have a compact volume of the freshwater after 2 years of injection. The use of three wells instead of one allows a reduction in the maximum overpressure in the injection cells from 2.6 m to about 1.1 m.

The fate of the injected wastewater can play an important role in water reuse because it may result in a further quality improvement due to the well-known phenomenon of Soil-Aquifer Treatment (SAT) (Idelovitch et al., 2003). Therefore, we have performed a last simulation in order to investigate the possibility of withdrawing the water previously injected into the limestone aquifer. With reference to the well layout implemented in scenario $S_{3A}$, a pumping well located in correspondence of the barycenter of the injection boreholes (× sign in Figure 13) is supposed to extract groundwater from the top portion of the aquifer starting after 2 years of water injection, i.e. from the final condition reached by scenario $S_{3A}$. Injection and withdrawal simultaneously occur for a 2-year period. The intake of the pumping well is 10 m long. The results are summarized in Figure 15, showing the evolution versus time of the salt concentration in the grid cell where the pumping well is located. After the first 2 years, when $c$ reduces from the initial groundwater value to approximately zero, the groundwater withdrawal at a rate equal to the cumulative injection rate ($Q_{withdrawn}=Q_{inj}=6'000$ m³/day) yields a certain deterioration of the groundwater quality in the well surroundings, and hence of the extracted water, with the salt concentration that rises to approximately 4 g/l, in steady-state conditions. Therefore, $Q_{withdrawn}$ must be reduced to get groundwater suitable for agricultural purposes. The numerical simulation points out that an almost fresh groundwater ($c<1$ g/l) can be permanently obtained by the system if the pumping rate is reduced to 2'000 m³/day. Notice that the injected TWW takes about 80 days to flow from the injection to the withdrawal intakes, which are about 100 m apart. Both the time period and distance are significant in terms of suspended solids filtration and organic matter biodegradation using secondary or tertiary wastewater effluents (Idelovitch et al., 2003; Kopchynski et al., 1996).



### 4.3. Model evaluation

Generally, the numerical models are simplifications of real aquifer systems. Model results are affected by (1) numerical approximations used to solve the ground water flow and transport equations, (2) discretization of the modeled area, and (3) the availability and accuracy of hydrogeological data used to define the spatial distribution of physical parameters, the boundary conditions, and the factors forcing the system (Idelovitch et al., 2003; Tsang, 2005). Another important limitation of calibrated numerical models is the non-uniqueness of their solutions.

The models presented here rely on a number of assumptions, starting from the hydrogeological structure of the system and the distribution of the hydrological properties ($K$, $S$, $\alpha_L$, and $\alpha_T$). The majority of the available stratigraphic information is located in the surrounding of the contemporary pond, with only little lithological data collected in the western desert zone. However, the results from the hydrogeophysical surveys carried out in the study site and the hydrogeological characterization carried out in other low-lying zones to the south, along the Wadi El-Natrun depression (Ammar, 2010; Khalil and Santos, 2013; Monteiro Santos and Sultan, 2008), have pointed out a remarkable lateral continuity and relative homogeneity of the main hydrological formations addressed by the modeling study.

Regarding the issue related to lithological and hydraulic heterogeneity, the two models did not address spatial variation of the hydrological parameters, namely the hydraulic conductivity, elastic storage, longitudinal and transversal dispersivity. Construction of the models in this manner would have required detailed hydrogeological information from aquifer tests and decadal piezometric records, which are not available. However, an accurate calibration of the depression-scale model is beyond the scope of the study. Indeed, its main objective was to evaluate if a groundwater flow model is able to reproduce the formation and growth of the pond as observed by satellite data implementing reasonable values of $K$ and $S$ (derived from the few pumping and recovery tests performed in the area) and estimate the factors forcing the system. The pond evolution over time allows for a global evaluation of the model representativeness of the hydrological processes in the area.

As reported above, the main assumption in the modeling of MAR is related to disregard of aquifer clogging in the surrounding of the injection wells. Deterioration of the aquifer properties due to biological, chemical, and physical clogging strongly depends on the quality of the water used in MAR (Bouwer, 2002). A number of methodologies have been proposed in literatures to account for the decrease of porosity and, thus, the conductivity of the medium due to clogging (Pérez-Paricio and Carrera, 2000). However, at the present stage of this study, the lack of specific information on the quality of the effluent from the Nubariya WWTP, and on the chemical and physical characteristics of the limestone composing the aquifer do not allow for a specific numerical investigation on the clogging effects.

Concerning the dispersivity values, a sensitivity analysis on $\alpha_L$ and $\alpha_T$ was carried out using the local model. The effect of the dispersivity values on the shape of the injected fresh-water volume was investigated to overcome the lack of dispersion and tracer tests. Due to the relatively small dimension of the local model (on the order of 2-3 km) and the homogeneity of the limestone aquifer suggested by the geophysical results, the assumption of parameter homogeneity seems more reliable here than in the depression-scale model.

### 5. Conclusions

In this study we evaluated the possible reuse of treated wastewaters in the Nubariya depression, Egypt, using extensive hydrogeophysical input and numerical modeling. The geophysical surveys successfully characterized the aquifer system in the study area where distinct layers of unsaturated sand, saturated sand and sandy clay were found. In addition, combined use of the MRS and TEM methods provided information about spatial distribution of the layers, as well as water quality and clay content. Within the sandy layers, we found that the water quality is inappropriate for use as drinking water or for agricultural purposes. In addition, a relatively shallow confining clay layer was found as one of the reasons for the evolution of the




artificial pond in the area. Below the clay layer, a 100m thick aquifer with high free water content was nominated as the potential aquifer for recharge.

The estimated hydrogeophysical model was used as an input for building the hydrological model of the area. The simulations carried out at the depression-scale clearly pointed out the inefficient use of the freshwater resources in the study area. Although not very well known, the problem of waterlogging and pond development within desert zones caused by

over-irrigation has been reported for several locations worldwide. This study investigates a phenomenon that represents a contradiction in areas where long-term programs are underway to effectively reuse treated wastewaters: large volumes of good-quality freshwater are wasted and large economic efforts are spent to improve and reuse TWW.

The same low-permeability layer, which is responsible for the formation of the artificial pond of Nubariya, confines a relatively thick limestone aquifer and bounds the upward migration of the treated waters injected into the system. The

continuity of the clay unit, an important requirement in view of the recharge, was clearly highlighted by the geophysical surveys. Although preliminary, a local-scale density-dependent groundwater flow and transport model allowed the development of an optimized MAR scheme.

This research highlights the hydrological challenges necessary for an effective management of water resources in reclaimed desert areas and highlight the effectiveness of using advanced geophysical and modeling methodologies to

characterize the subsurface environment, investigate naturally and/or anthropogenically driven exchanges between groundwater and surface water, and plan appropriate interventions aimed at efficient use of available waters.

## 6. Acknowledgments

This article is an outcome of the IMPROWARE Project funded by the European Union in the framework of the SWIM (Sustainable Water Integrated Management) Programme. Ahmad A. Behroozmand was also supported by funding from The

Danish Council for Independent Research – Natural Sciences. The authors are much indebted to Ehab El-Hemady and Amr Fadlelmawla, Egyptian Environmental Affairs Agency, for the essential support to the field surveys in Egypt and to Andrea De Angelis, Italian Ministry for the Environment, Land and Sea, for his effort in managing the IMPROWARE activities. We are extremely grateful to Simon Rejkjaer, Mads S. Christensen and Henrik Boejer who in these so challenging times of 2013 and 2014 collected the geophysical data together with a great crew from Egypt, particularly Sayed Bedair and Ibrahim Yehia

Ibrahim. Interested readers can access the data by contacting Ahmad A. Behroozmand at abehrooz@stanford.edu.

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



**Tables**

**Table 1. Chemical analyses of groundwater and pond water samples collected in 2003 (after *Sharaky et al.* (2008)), 2011 (after *Masoud* (2014)), and 2014 within the IMPROWARE project\*.**

| Parameter | Groundwater | | | Pond Water |
|---|---|---|---|---|
| | 2003 | 2011 | 2014 | 2014 |
| $E_c$ (mS/cm) | 2.8 – 3.9 | n.a. | 25.5 – 37.8 | 26.0 |
| TDS (ppm) | 1'860 – 2'550 | 2'000 – 4'000 | 16'300 –20'300 | 17'150 |
| pH (-) | 7.7 – 8.6 | 6.7 – 8.6 | 6.9 – 7.7 | 8.0 |
| $Na^+$ (mg/l) | 590 – 690 | 680 – 1'600 | 5'300 – 7'300 | 5'400 |
| $Cl^-$ (mg/l) | 760 – 1'070 | 900 – 3'000 | 7'700 – 11'100 | 7'800 |
| $Fe^{2+}$ (mg/l) | 0.6 – 3.0 | 0.5 – 2.2 | <0.1 | <0.1 |
| $Mn^{2+}$ (mg/l) | 0.1 – 0.3 | 0.1 – 0.2 | <0.1 | <0.1 |
| $Zn^{2+}$ (mg/l) | 0.1 – 1.2 | 0.3 – 0.6 | 0.3 – 0.7 | <0.1 |
| $Cu^{2+}$ (mg/l) | <0.1 | <0.1 | <0.1 | 0.1 |
| $Ni^{2+}$ (mg/l) | 0.1 – 0.2 | 0.1 – 0.2 | n.a. | n.a. |
| $NO_3^-$ (mg/l) | 2.5 – 6.0 | 3.0 – 8.0 | 0.2 | 0.2 |

\* $E_c$: Electrical conductivity; TDS : total dissolved solids





**Table 2. WalkTEM system setup.**

| 40x40 m system | Low Moment | High Moment |
| --- | --- | --- |
| No. of turns | 1 | 1 |
| Area | 1'600 m$^2$ | 1'600 m$^2$ |
| Current | ~ 1.1 A | ~ 12 A |
| Tx Moment | ~ 1'760 Am$^2$ | ~ 19'200 Am$^2$ |
| Waveform | Square | Square |
| First gate | 5.6 μs | 9.0 μs |
| Last gate | 0.9 ms | 8.8 ms |
| **100x100 m system** | **Low Moment** | **High Moment** |
| No. of turns | 1 | 1 |
| Area | 10'000 m$^2$ | 10'000 m$^2$ |
| Current | ~ 2.1 A | ~ 23 A |
| Tx Moment | ~ 21'000 Am$^2$ | ~ 230'000 Am$^2$ |
| Waveform | Square | Square |
| Fist gate | 9.2 μs | 47 μs |
| Last gate | 2.2 ms | 21 ms |




**Table 3. MRS measurement configuration used in this study.**

| Parameter | Value |
| --- | --- |
| Tx/Rx loop side length | 100 m |
| No. of wire turns | 1 |
| No. of pulse moment | 16 |
| No. of stacks | 30 |
| Noise acq. time (before FID) | 1 s |
| Energizing pulse duration | 30 ms |
| Instrument dead time | 20 ms |
| Recording time (FID) | 1 s |






**Figures**

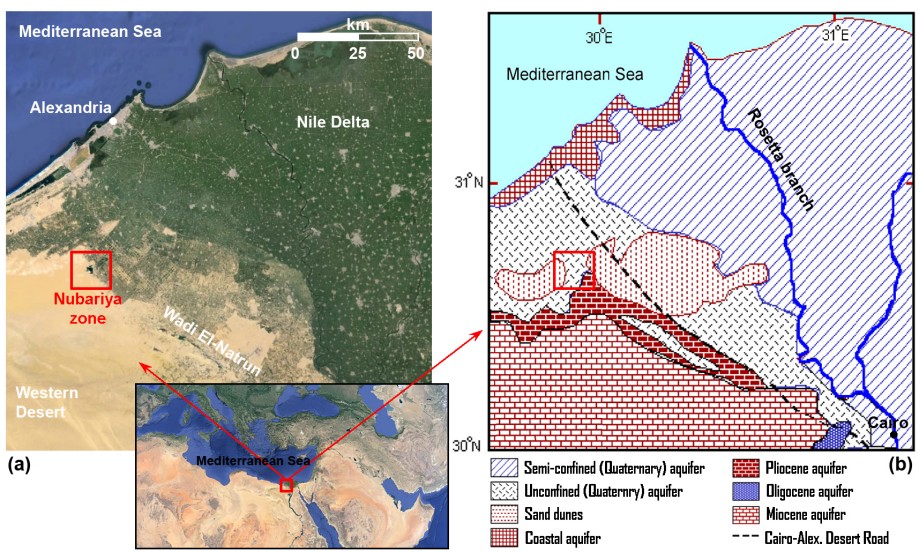

**Figure 1. (a) Location map of the Nubariya depression on the margin between the Western Desert and the western Nile Delta, Egypt. (b) Map showing the main hydrogeological and aquifer units (modified after *Sharaky et al.* (2008)).**






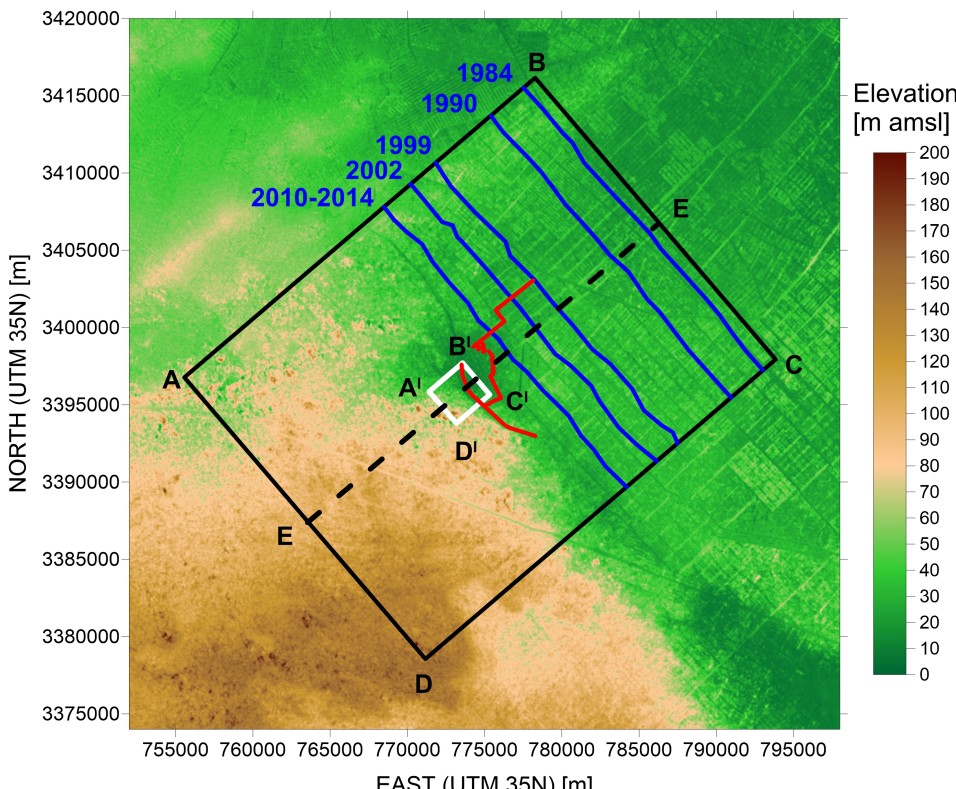

**Figure 2. Digital Elevation Model of the Nubariya depression and its surroundings derived from the ASTER-GDEM (Advanced Spaceborne Thermal Emission and reflection Radiometer and Global Digital Elevation Model) (Tachikawa et al., 2011) and calibrated by a kinematic DGPS in situ survey (red alignment). The boundaries of regional and local models developed in the framework of the IMPROWARE Project are shown by the black and white boxes, respectively. The blue lines and associated dates, which separate the desert (to the southwest) from the irrigated areas (to the northeast), indicate how the latter has encroached south-westwards over time. This information was obtained form an analysis of Landsat images. The results from the 3D FE pond-scale groundwater flow model are shown along the black alignment E-E.**







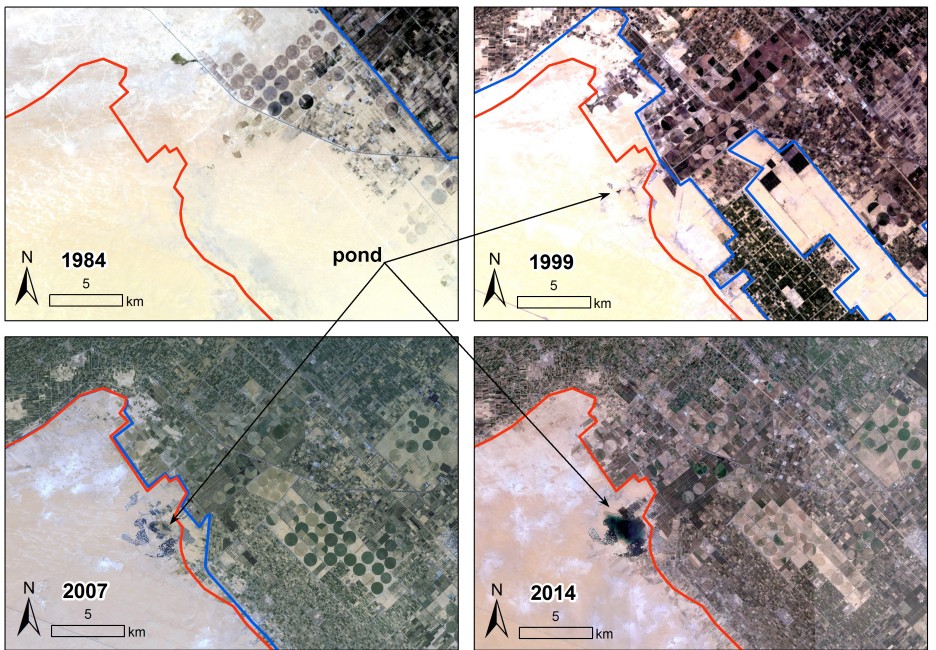

**Figure 3. Landsat images showing the time evolution of the farmland (blue lines) and the pond in Nubariya between 1984 and 2014. The extent of reclamation of the desert is illustrated with reference to the red line which marks the contemporary (2014) extent of the irrigated area.**





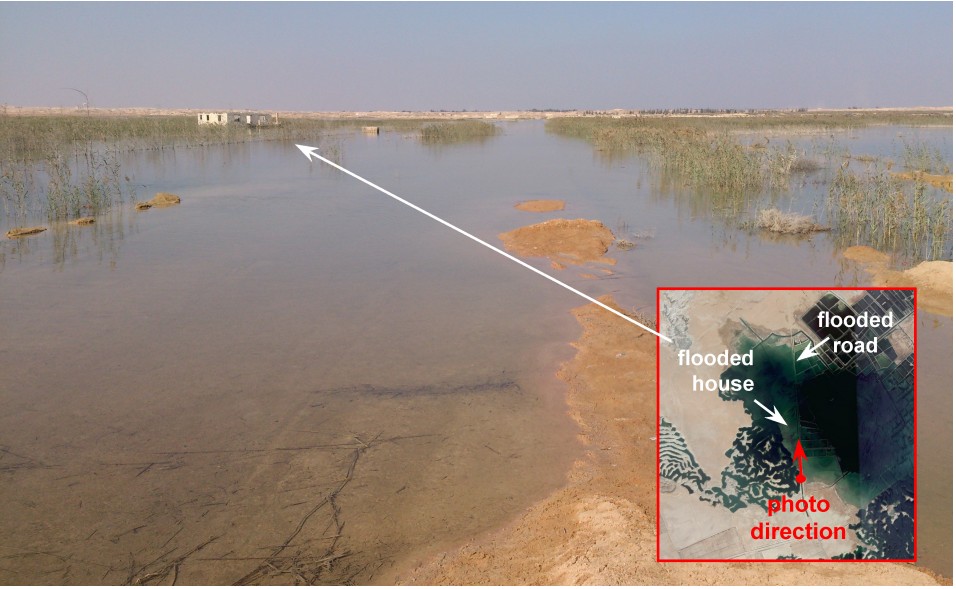


**Figure 4. Photo of the Nubariya pond dated November 2013 showing houses and road abandoned due to the pond expansion. The photo location and direction is shown in the inset.**





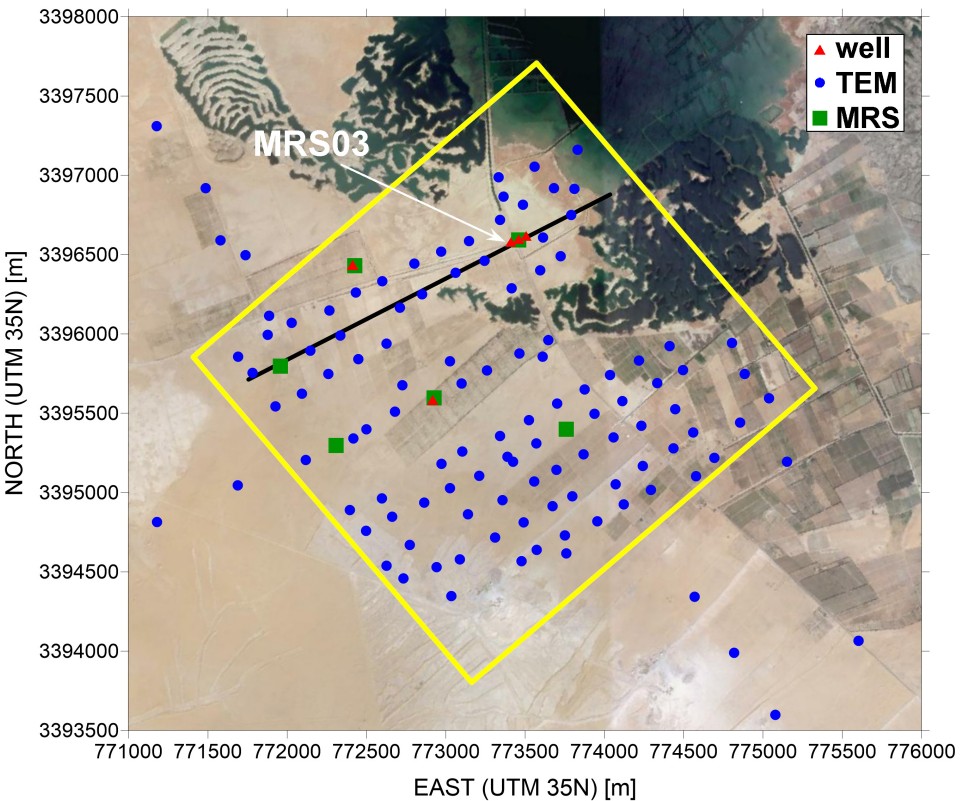

**Figure 5.** Location map of the boreholes (red triangles), TEM (blue dots) and MRS (green squares) measurements carried out in the surroundings of the Nubariya pond. The black line represents the trace of the cross section through the resistivity model shown in Figure 6. The location of MRS03 is highlighted. The domain of the local-scale model is shown by the yellow box. The background is a Landsat image acquired in 2014.





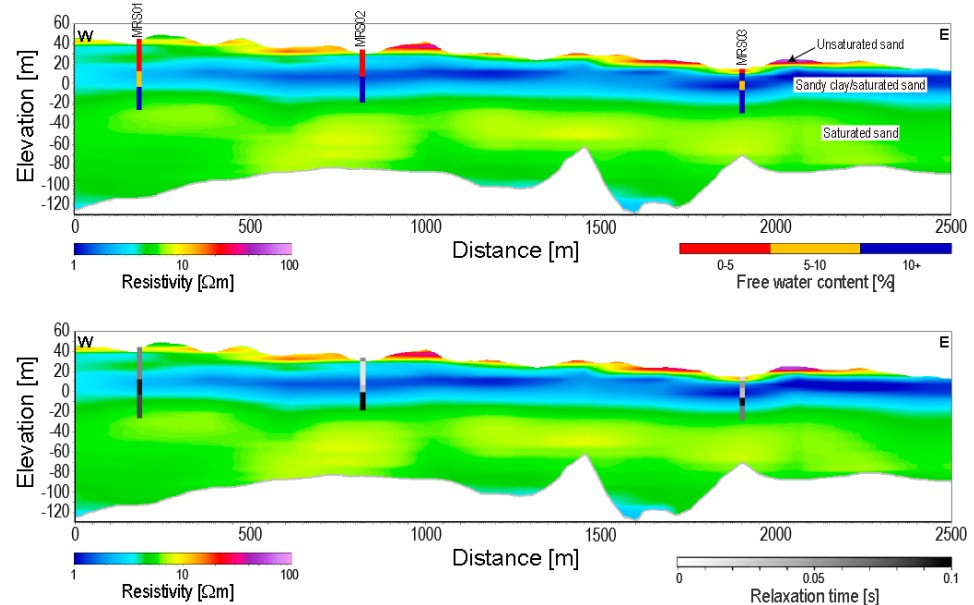


**Figure 6. A west-east section of the resistivity model obtained from the TEM data along the black line in Figure 5. The color is faded white below the depth of investigation (gray line). The bars on the plots represent three MRS results in the vicinity of the profile in terms of the free water content (top) and relaxation times (bottom). Figure 5 provides the location of the MRS soundings.**






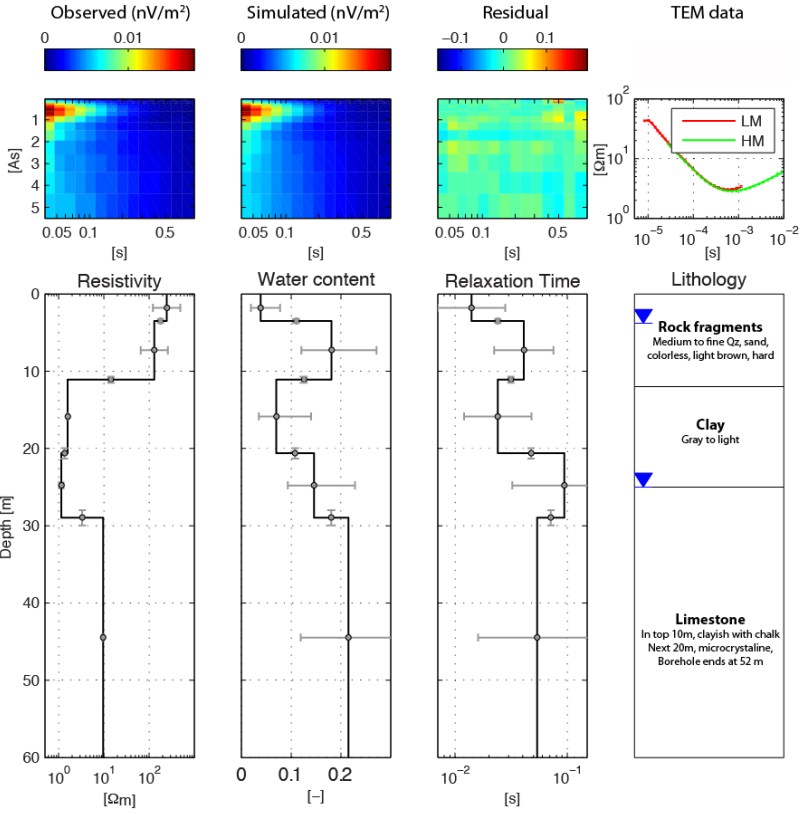

**Figure 7. Inversion results for sounding MRS03 (see Figure 5 for location of the sounding). The MRS and TEM data were inverted jointly. Row1: the MRS observed (column 1) and simulated (column 2) data, and weighted data residuals (column 3). Column 4 shows the TEM data fit. Row 2: the inversion results (black lines) in terms of resistivity, free water content and relaxation time (columns 1-3, respectively), together with model parameter uncertainties (gray error bars). Column 4 shows lithological information obtained from a nearby borehole.**






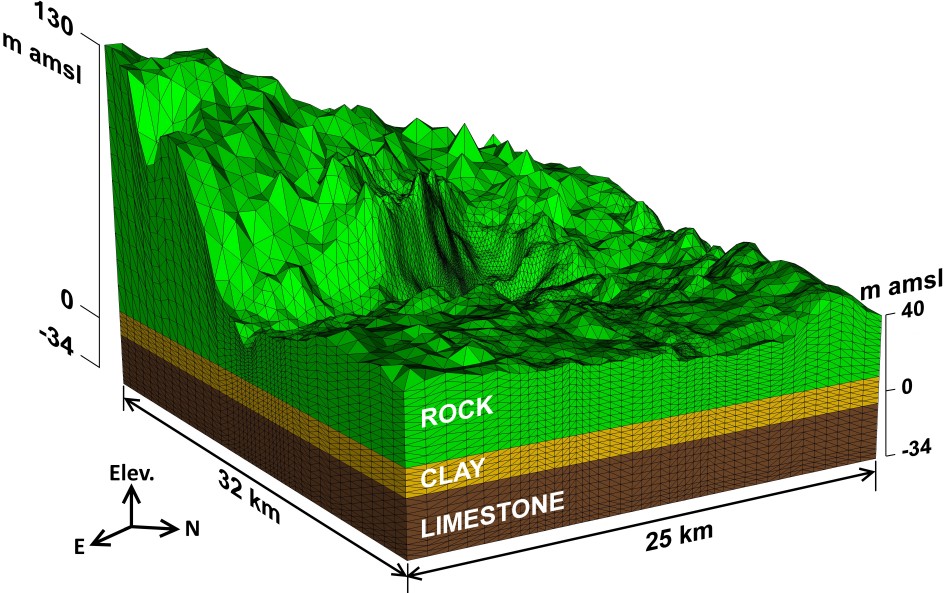

**Figure 8. 3D mesh of the regional hydrogeological model. A 2D triangular grid was projected vertically to generate the 3D FE tetrahedral mesh used in the CATHY simulations.**






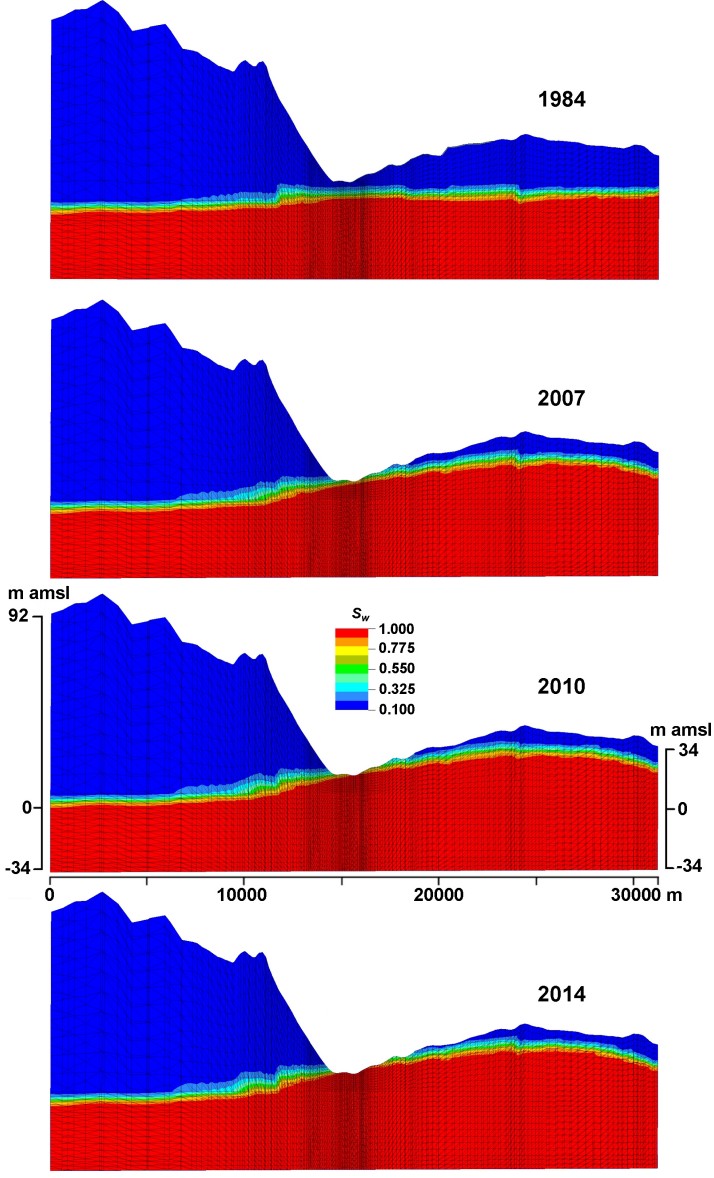

**Figure 9. Evolution of the saturation degree along the vertical cross-section E-E traced in Figure 2, i.e. along the main groundwater flow direction, as obtained by the 3D depression-scale model.**






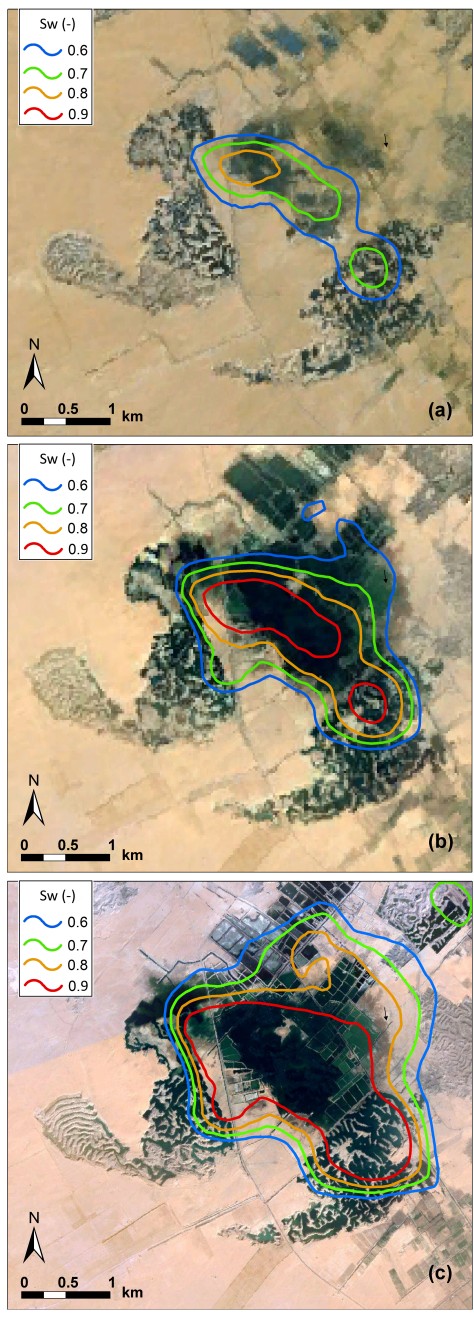

**Figure 10. Contour lines of the saturation degree at the land surface around the depression in (a) 2007, (b) 2010, and (c) 2014 as obtained by the 3D pond-scale model. The background is Landsat images acquired in 2007, 2010, and 2014, respectively.**





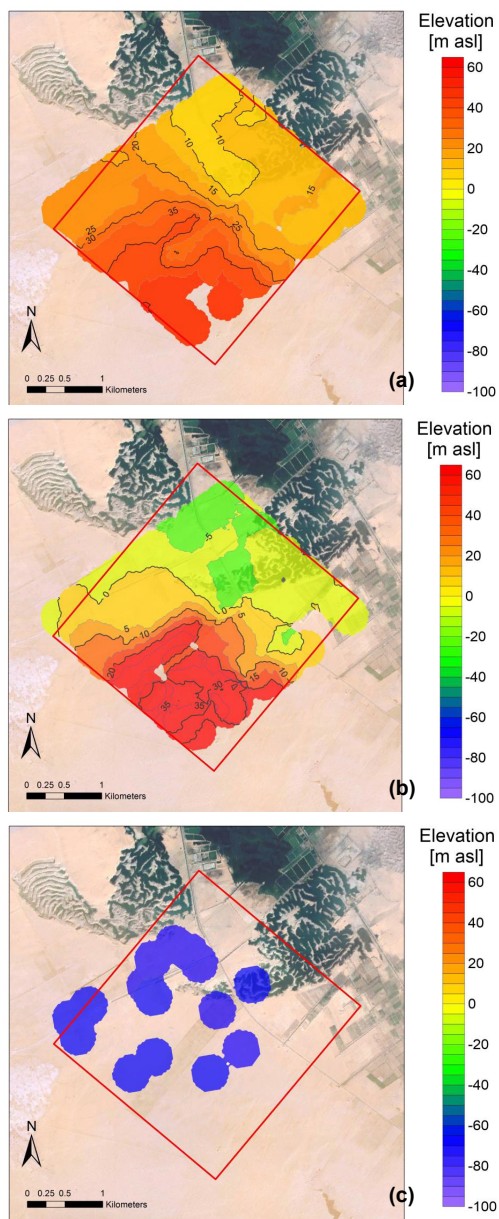

00

**Figure 11. Elevation maps (m amsl) of (a) top of the clay unit, (b) bottom of the clay unit, i.e. top of the limestone confined aquifer, and (c) bottom of the limestone aquifer as provided by the hydrogeophysical (deep TEM) surveys and implemented in the local FE model.**


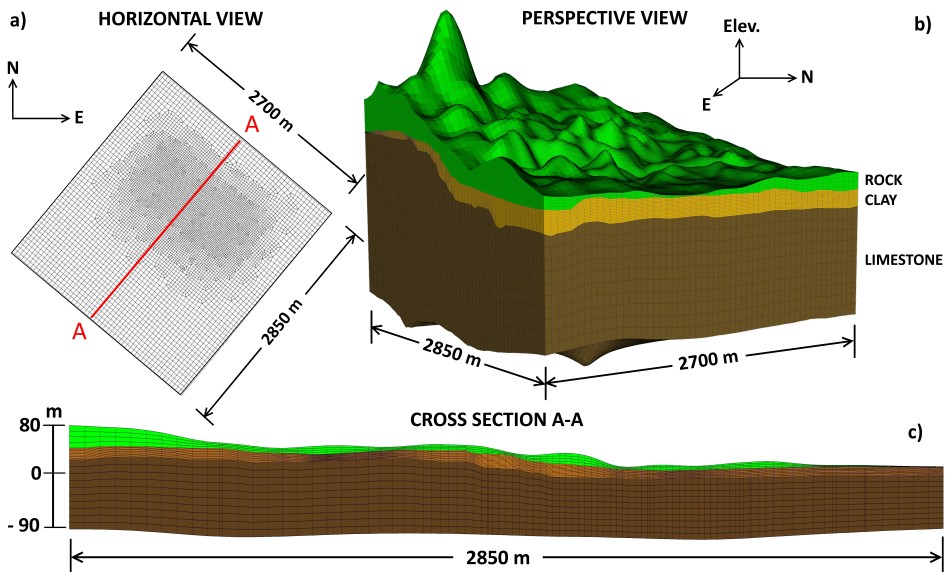

05

**Figure 12. 3D FE grid used in SUTRA to simulate possible scenarios of artificial aquifer recharge at Nubariya: (a) horizontal view, (b) perspective view, and (c) vertical cross-section along the A-A alignment traced in (a).**





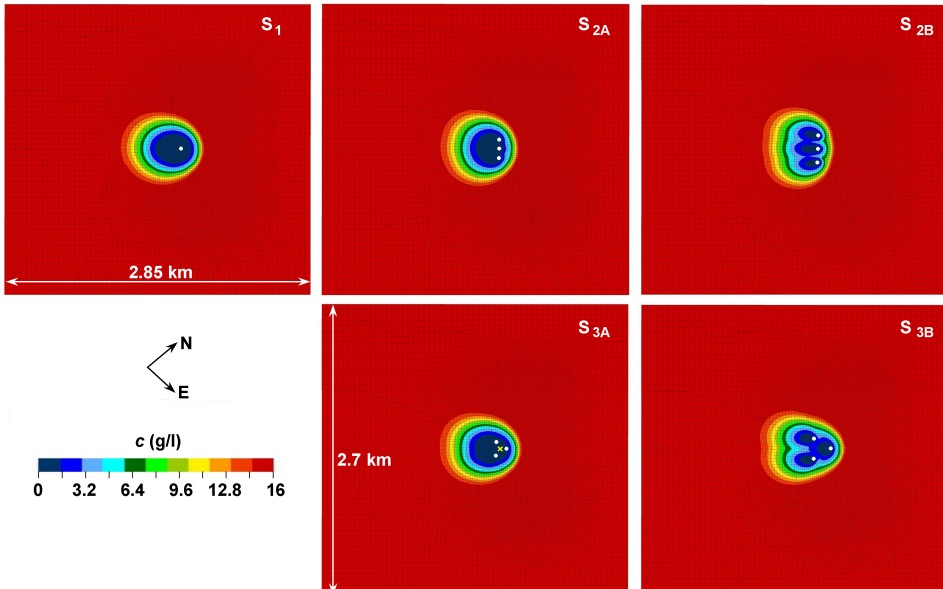

**Figure 13. Salt concentration (g/l) on a horizontal plane at -30 m amsl (i.e., at the top of the injected aquifer) after 2 years of aquifer recharge for the various scenarios addressed by the study. The location of the injection wells are shown by white dots. The production well in scenario S$_{3A}$ is marked by a yellow X mark.**

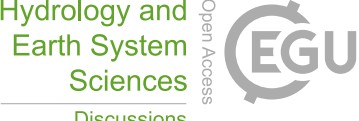

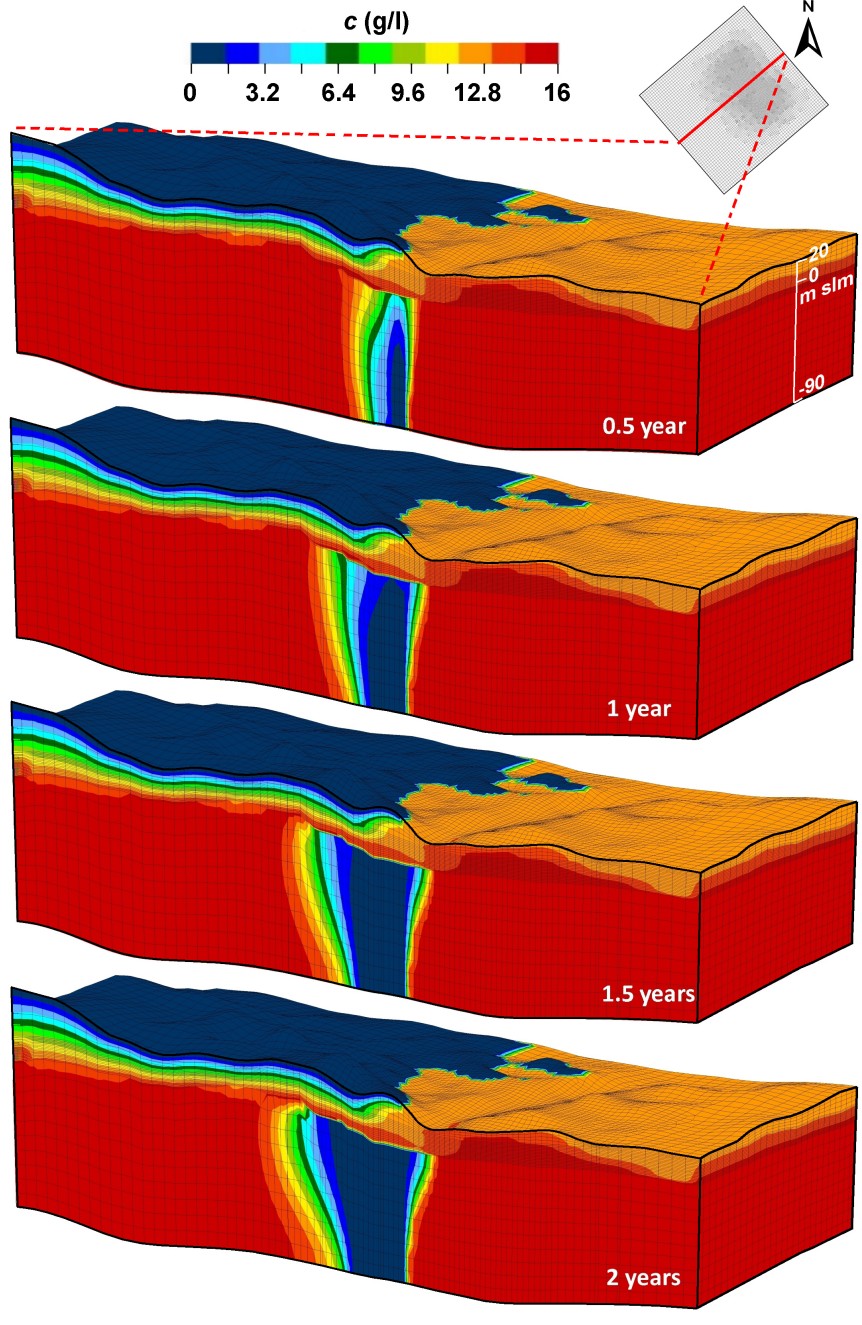

15    **Figure 14. Scenario S$_{3A}$: evolution of the salt concentration (g/l) on a vertical section through the barycenter of the injection configuration. Vertical exaggeration is 10.**





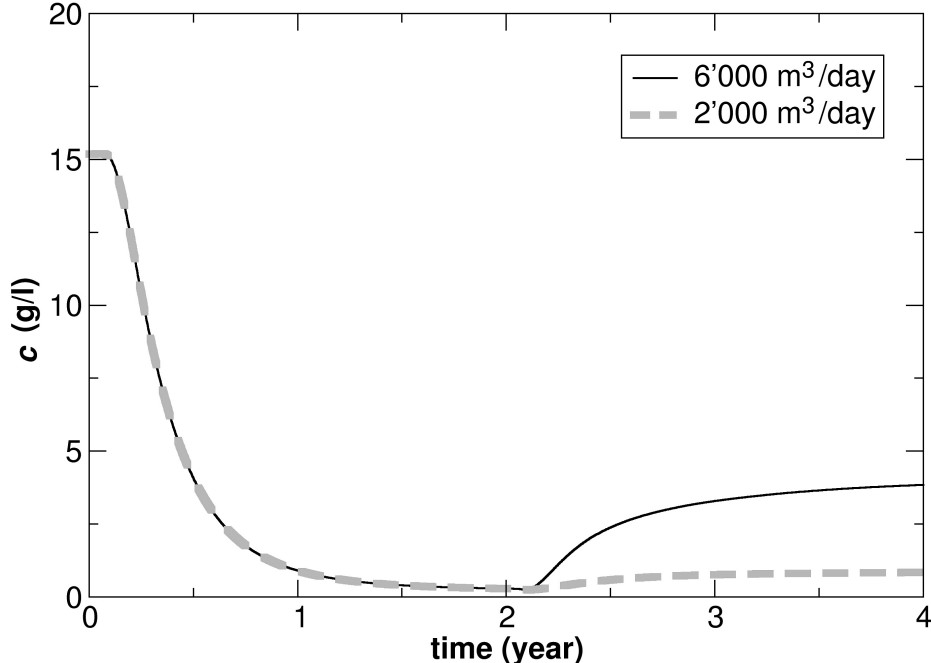

**Figure 15. Salt concentration versus time at the pumping well intake for the two selected values of $Q_{withdrawn}$.**