# Peer review of "Anthropogenic wetlands due to over-irrigation of desert areas; A challenging hydrogeological investigation with extensive geophysical input from TEM and MRS measurements"

_Hydrology and Earth System Sciences, 2016_

## Referee Comment (RC1) · L. Tosi (Referee) · 20 Jan 2017

The MS "Anthropogenic wetlands due to over-irrigation of desert areas; A challenging hydrogeological investigation with extensive geophysical input from TEM and MRS measurements" by Behroozmand et al. addresses a topic that is of critical importance for improving (i) the knowledge on the surficial water – groundwater interaction in "arid wetlands" and (ii) the effective management of water resources in reclaimed desert areas. The authors show an original approach based on the integration of magnetic resonance sounding and ground-based transient EM survey outcomes. Geophysical data, calibrated by hydrogeological (e.g., continuous core sampling, pumping and recovery tests) investigations and chemical analyses, allowed to delineate the main hydro-stratigraphic units and characterize the groundwater quality of the upper 100m subsoil setting. Remote sensing investigation (i.e. DEM and satellite images) completed the assessment of the hydrological and geomorphological setting of the study area. All these information produced the input data for building a regional groundwater flow model, a local density-dependent flow model and a transport numerical model used to reconstruct the evolution of the aquifer system (i) and to develop scenarios of artificial aquifer recharge using the treated waters (ii). The simulations of the hydrological response of the lowlands to the reclamation activity carried out in the 1984 – 2014 period point out the surplus of irrigation water as responsible for the formation and the growth of the pond as well as the inefficient use of the freshwater resources. Finally, the local-scale density-dependent groundwater flow and transport model provide key information on the possibility to reuse treated wastewater for aquifer recharge. The general concept of the study is interesting and really nice and the MS presents novel data and results. The experimental application is well-designed and described, and it could be widely applicable in various "arid wetlands". The MS is well written, pleasant to read and of interest for a broad scientific community. The introduction gives a good overview on the relevant questions and properly motivates the goal of the study. Results are well supported by data.

I congratulate the authors for this nice scientific contribute. I would strongly recommend the MS for publication once the following comments are address. None of them affect the overall quality of the MS.

P2_L69 -75. I do think it would be useful to improve this section. This is just to avoid possible misunderstanding. It is not clear which parts of this section refer to "natural wetlands" or "artificial wetlands. For instance, P2_L68-69: ". . . can evolve into anthropogenic perennial in-land wetlands even in arid or semi-arid regions. These "Arid wetlands", i.e. natural humid zones in an arid or semiarid climate, . . .". If the authors

refer the "Arid wetlands" to the anthropogenic perennial in-land wetlands, I suggest to change "natural humid zones" with "man-induced humid zones" (or artificial humid zones). Otherwise, I suggest: The "arid wetlands" . . . . . ..

P2_L72-73. Similarly, ". . . Worldwide arid wetlands are threatened by increasing anthropogenic pressure. . ..". What do the authors mean with Worldwide arid wetlands? Are both natural and artificial arid wetlands? It is not clear whether the arid wetlands are only those induced by man.

P3_L05-06. ". . . coastal areas through re-injection of treated water - http://www.improware.eu/) project aimed at evaluating the possible reuse of treated wastewaters (TWW) in this area. . ." I recommend the authors to better specify the connection between the focus of the paper (i.e. P3_L96-03) and the re-injection of treated water. For instance, is the "treated water" used for reinjection from the surface water accumulation?

P4_51-52. What do the authors mean with "the climatic conditions" (e.g., temperature and humidity)?

P4_52-53. I recommend to add at least one reference on the "Hydrogeological investigations at the regional scale".

P4_53. As "regional scale" has been considered, what do the authors mean with the upper 200m? Do they refer to the whole area shown in Figure 2 or only the lowlands sector?

P4_52-55. This part is not adequately clear: "Hydrogeological investigations at the regional scale highlight the presence of loose coarse Miocene sands with clay lenses in the upper 200 m overlain by Pliocene. . ." I suggest: Hydrogeological investigations at the regional scale highlight the presence of loose coarse Miocene sands with clay lenses overlain by Pliocene deposits in the upper 200 m of subsoil. The Pliocene units consist mainly of estuarine clayey facies at the base, passing upward to fluvio-marine
and shallow marine limestones. The uppermost units are exposed in the lowest parts of the landscape and their vicinities.

P4_L56 "...unit...", perhaps, units or deposits?

P5_L65. What do the authors mean with "contemporary"? Perhaps you mean "recent" or 2014 (i.e. contemporary the acquisition time of DGPS data).

P5_L81. Check "Error! Reference source not found"

P5_93. Delete "red triangles in" as it is already reported in the figure caption.

P5_01. Check "Error! Reference source not found"

P6_26. Check "Error! Reference source not found".

P7_71. Could the authors confirm that the climate conditions were reasonably similar during WalkTEM and MRS surveys, i.e. December 2013 and January 2014, respectively?

P7_L77&80. Check "Error! Reference source not found".

P8_11. Overall, the...

P9_L39. I suggest to delete "Contemporary".

P9_L47. I suggest to use "retrieved or "obtained" instead of defined.

P11_L5. Delete "Within the framework of IMPROWARE" (it is already mentioned in the previous lines)

P12_L56-57. ...lowest portion of the limestone... I suggest to add the depth range of the borehole intakes.

P19_Table 1. Groundwater and Pond Water vs 2003, 2011, 2014 are not clear.

P22_Fig.1. Satellite images source is missing.

P24_Fig.3. Satellite images source is missing.

P25_Fig.4. Satellite image source is missing (the inset).

P26_Fig.5. Satellite image source is missing.

P31_Fig.10. Satellite images source is missing.

P32_Fig.11. Satellite images source is missing.

---

## Author Comment (AC1) · 6 Feb 2017

Dear Prof. Fan,

We thank you and reviewer Dr. Luigi Tosi for evaluation of our paper. Following the reviewer recommendations and comments, we have carefully revised the manuscript. Attached please find our point-by-point response letter to the reviewer and a revised manuscript (with modifications marked). The figures are intact with respect to the original manuscript.

Kind regards, Ahmad A. Behroozmand - on behalf of all co-authors

Please also note the supplement to this comment:
http://www.hydrol-earth-syst-sci-discuss.net/hess-2016-630/hess-2016-630-AC1-supplement.zip

———————————————————

---

## Referee Comment (RC2) · Anonymous Referee #2 · 18 Feb 2017

General comments

Dear Authors, This article address the use of hydrogeophysics for characterizing an anthropogenic wetland, and hydrogeological modeling to study scenarii of remediation. The text is well written, accessible thanks to a short introduction to each method and references for more details.

fig 9 and 10 are well demonstrative: the model built on hydrogeophysics data fits very well the observed pond evolution.

I particularly appreciate the clarity of the results (especially fig 10, 15) and the discussion on the limits of the modeling results in paragraph 4.3. The clogging in the vicinity of the injection wells..

It would have been valuable, in my opinion, to add in the discussion the impact of the quality / accuracy of the hydrogophysical model on the hydrogeological modeling. Here, TEM data and MRS are of very good quality, and the model is not very heterogeneous: the low lateral variation result in the use of almost a layered aquifer model. In a more complex case, the density of measurement should have been increased ?

My review is based on the revised_manuscript version after the response of authors to the 1st reviewer, and I notice no spell errors or reference missing.

I recommand this article for publication as it is.

few remarks in the text:

fig 6 and paragraph 3.2.2 Is the conductive 4th layer really a robust result of the inversion of TEM data ? proved by boreholes? (I know authors are expert in the use of TEM)

If not, this deep layer is used in the modelling as a bottom condition, far below the pumpings..., I d'ont think it affect the results.

The same question arises in p8 line 00.. I suppose fig 11 c provides answer to my inquiry. Perhaps a short discussion on it and a reference to fig 11 earlier in the text would clarify this point (in paragraph 3.2.2 for instance).

fig 6. MRS colorscale for decay time.. is not really demonstrative..

p7 - 3.3 line 65 - you wrote: "based on their relaxation time values, the soils [pore diameter or grain size...] can be classified as fine, medium and large materials." rephrase (or remove this sentence, as it is clearly explained just after) ... MRS decay time characterizes pores... not grain... even if grain size and pore size are linked in sand / sandstone rocks

---

## Author Response (AR1)

General comments
Dear Authors, This article address the use of hydrogeophysics for characterizing an anthropogenic wetland, and hydrogeological modeling to study scenario of remediation. The text is well written, accessible thanks to a short introduction to each method and references for more details.
fig 9 and 10 are well demonstrative: the model built on hydrogeophysics data fits very well the observed pond evolution.
I particularly appreciate the clarity of the results (especially fig 10, 15) and the discussion on the limits of the modeling results in paragraph 4.3. The clogging in the vicinity of the injection wells.
We thank the reviewer for positive evaluation of our work.

It would have been valuable, in my opinion, to add in the discussion the impact of the quality / accuracy of the hydrogophysical model on the hydrogeological modeling. Here, TEM data and MRS are of very good quality, and the model is not very heterogeneous: the low lateral variation result in the use of almost a layered aquifer model. In a more complex case, the density of measurement should have been increased?
We thank the reviewer for this comment. We added this description in section 4.3.

My review is based on the revised_manuscript version after the response of authors to the 1st reviewer, and I notice no spell errors or reference missing.
I recommend this article for publication as it is.

few remarks in the text:

fig 6 and paragraph 3.2.2 Is the conductive 4th layer really a robust result of the inversion of TEM data ? proved by boreholes? (I know authors are expert in the use of TEM)

If not, this deep layer is used in the modelling as a bottom condition, far below the pumpings..., I d'ont think it affect the results.
The same question arises in p8 line 00. I suppose fig 11 c provides answer to my inquiry. Perhaps a short discussion on it and a reference to fig 11 earlier in the text would clarify this point (in paragraph 3.2.2 for instance).
The results presented in Figures 6 and 11 are robust and reliable. For each TEM data, the resistivity model is plotted down to a depth where enough sensitivity exists (depth of investigation, DOI). As shown in figures 6 and 11, and as stated in Section 3.2.2 (page 6, line 39), different DOI values are calculated for each TEM sounding that depends on total sensitivity of acquired data on each specific model. Therefore, the information regarding the 4$^{th}$ layer, wherever obtained, is reliable.

fig 6. MRS colorscale for decay time.. is not really demonstrative..
We increased resolution of the figure.

p7 - 3.3 line 65 - you wrote: "based on their relaxation time values, the soils [pore diameter or grain size...] can be classified as fine, medium and large materials." rephrase (or remove this sentence, as it is clearly explained just after) ... MRS decay time characterizes pores... not grain... even if grain size and pore size are linked in sand / sandstone rocks
We deleted this sentence.

**List of all relevant changes in the manuscript**

A short description relevant to the reviewer's comment is added in Section 4.3.

[revised manuscript text omitted]